# Understanding the Protective Effect of Liquid Nitrogen Freezing on Crayfish Quality During Transportation and Storage

**DOI:** 10.3390/foods14122078

**Published:** 2025-06-12

**Authors:** Gehao Lei, Peng Zhang, Limin Xu, Liuqing Wang, Xiaoyue He, Jiwang Chen

**Affiliations:** 1College of Food Science and Engineering, Wuhan Polytechnic University, Wuhan 430023, China; lghwhpu2024@163.com (G.L.); xlmwhpu@163.com (L.X.); wlq1015@whpu.edu.cn (L.W.); hxiaoyue2024@163.com (X.H.); 2Hubei Key Laboratory for Processing and Transformation of Agricultural Products, Wuhan Polytechnic University, Wuhan 430023, China

**Keywords:** crayfish, liquid nitrogen freezing, freeze–thaw cycles, myofibrillar protein, functional properties

## Abstract

Freezing has been widely used to preserve the freshness and quality of crayfish (*Procambarus clarkii*). However, temperature fluctuations during transportation and storage inevitably affect the quality attributes of crayfish. In this study, the effect of liquid nitrogen freezing (LNF) on crayfish myofibrillar protein (MP) was investigated under freeze–thaw (FT) cycles. The small ice crystals formed by LNF could reduce the conversion of sulfhydryl groups to disulfide bonds, preventing the exposure of hydrophobic groups, thereby maintaining the functional properties of MP. LNF could prevent the degradation and oxidation of MP and maintain its compact and smooth microstructure. Compared to refrigerator freezing (RF), LNF showed a stronger protective effect on the secondary and tertiary structures of MP, alleviating their conformational changes. Therefore, LNF could be an alternative freezing method to preserve crayfish quality against FT cycles during transportation and storage.

## 1. Introduction

Crayfish (*Procambarus clarkii*), well known for its special flavor, economic efficiency, high edible value and delicious taste, has become an important economic breeding variety in China [1]. The cultivation amount and the breeding area of crayfish in 2023 were 3.16 million tons and 29.5 million acres, respectively [2]. However, crayfish industry development and market penetration could face challenges stemming from the inherent seasonality and geographically concentrated nature of this production [3]. Freezing is a common method to maintain the quality of crayfish. Conventional freezing techniques like air and plate freezing often compromise crayfish quality owing to the prolonged freezing duration and their propensity to generate sizable ice crystals [4]. Liquid nitrogen freezing (LNF) utilizes the extremely low temperature of liquid nitrogen to quickly freeze food. It has advantages like excellent preservation ability, long shelf-life extension, low energy consumption, small footprint, wide applicability and environmental friendliness in food preservation. However, it also has disadvantages such as high cost, potential promotion of food damage, high equipment requirements and high storage and transportation requirements, which limit its application in low-value products. For some high-value products, such as crayfish, a more rapid freezing speed contributes to preventing the formation of large ice crystals, which is an advantage for maintaining crayfish quality [5].

With the prolongation of the freezing storage time, the deterioration of the biochemical and physical properties of crayfish meat influences the quality attributes of crayfish products, which finally shortens their shelf-life. Crayfish frozen by LNF holds texture quality and muscle fibers better, resulting from less protein denaturation and cell rupture [6]. As a frontier refrigeration technology for aquatic products, LNF achieves enhanced crayfish quality through controlled ice crystallization processes that prevent cellular membrane disruption [7]. Nevertheless, due to the time required to market these frozen crayfish, temperature fluctuations (−18 °C to −5 °C) inevitably occur during transportation and storage [8]. Myofibrillar protein (MP) plays an important role in the quality of crayfish, because the interaction between actin and myosin maintains the shape and integrity of crayfish muscle. Repeated FT cycles accelerate crystalline lattice reorganization in crayfish, manifesting as ice crystal growth amplification. This phenomenon could destroy the muscular tissue structure and the conformation of MP, leading to protein degradation and water loss [9]. Crayfish quality degradation is generally caused by damage to muscle fibers resulting from ice crystallization and the freezing-induced denaturation of MP, which further affects crayfish color, flavor and taste [10,11]. Few studies have focused on crayfish preservation by LNF during transportation and storage. In our previous study, we explored the effect of LNF on crayfish quality by examining crayfish macro muscle structure during FT cycles [12]. However, the mechanisms of LNF protection of MP conformation, microstructure and functional properties are still unclear.

This study delves into how LNF affects the functional properties, microstructure, protein integrity and conformation of frozen crayfish MP during FT cycles. By analyzing changes in the molecular structure of MP, it reveals mechanism by which LNF protects frozen crayfish quality. These findings offer a theoretical foundation for aquatic product cryopreservation.

## 2. Materials and Methods

### 2.1. Materials

Crayfish and liquid nitrogen were friendly provided by Hubei Qianwang Ecological Crayfish Industrial Park Group Co., Ltd. (Qianjiang, China), and Hubei He yuan Gas Co., Ltd. (Yichang, China), respectively. Piperazine-N,N′-bis (PIPES), phosphate-buffered saline (PBS) and 2-nitrobenzoic acid (DTNB) were purchased from Yuan ye Biotechnology Co., Ltd. (Shanghai, China). Potassium chloride, sodium hydroxide, potassium sulfite, hydrochloric acid, potassium sodium tartrate tetrahydrate, copper sulfate pentahydrate, urea, potassium bromide, sodium dihydrogen phosphate and disodium hydrogen phosphate (all were analytically pure) were purchased from China National Pharmaceutical Group Chemical Reagent Co., Ltd. (Shanghai, China).

### 2.2. Preparation of Crayfish

Live crayfish (similar in size, bright red in color, and weighing 20.0 ± 5.00 g) were frozen in foam boxes with ice packs and transported to the laboratory within 2 h. Crayfish of good quality were selected as experimental samples. Crayfish without freezing treatment were designated as the non-frozen group (NF). The treated crayfish were randomly divided into three groups, frozen at −20 °C, −50 °C (RF) and −80 °C (LNF) and transferred to a −18 °C freezer for storage. After 45 days of preservation, crayfish was thawed at 4 °C (this process was defined as one FT cycle). Crayfish tails after 1–5 FT cycles were taken and shelled, and the intestines were removed. These crayfish tails were labeled as n FT, with n indicating the number of FT cycles.

### 2.3. Extraction of Crayfish MP

Crayfish MP was extracted following the method of Park et al. [13]. Crayfish muscle was homogenized with a 1:4 (*w*/*v*) phosphate solution (50 mL of PBS containing 2 mL of MgCl_2_, 0.1 M NaCl and 1 mL of EDTA, pH 7.0). After centrifugation, the precipitate was washed twice with PBS and then with 0.1 mL of NaCl buffer (in 50 mL of PBS, pH 7.0). The final precipitate was MP.

### 2.4. Methods for Determining the Functional Properties of MP

#### 2.4.1. Solubility

The measurement of solubility followed the method of Chen et al. [14]. Protein concentration was set to 1 mg/mL with phosphate buffer, and the solution was homogenized by agitation to ensure the uniform dispersion of MP. Subsequently, the mixed MP solution was centrifuged. The resulting supernatant was harvested, and the total protein levels were quantified through application of the Biuret colorimetric assay. The solubility of MP was determined based on Equation (1).(1)$$\mathrm{Solubility}\;\left(\%\right)=\frac{A}{B}\times 100$$
where *A* represents the protein concentration of the supernatant, in mg/mL; and *B* represents the original protein concentration, in mg/mL.

#### 2.4.2. Turbidity

The turbidity of MP was measured using a UV-vis spectrophotometer (UV-1800, Shimadzu, Japan) according to the method of Chelh et al. [15]. Absorbance measurements of the MP solution (1 mg/mL) were performed at 660 nm under controlled temperature conditions (30, 40, 50, 60, 70 and 80 °C).

#### 2.4.3. Foaming Properties and Foaming Stability Determination

Foaming properties and foaming stability were analyzed following the method of Sharifian et al. [16] with slight modifications. The MP solution was diluted to 2 mg/mL and homogenized (12,000 rpm, 1 min). The homogenate was immediately transferred to a graduated cylinder, and the total volume was immediately measured, which was the sum of the foam volume and the liquid volume right after homogenization, denoted as *V*_0_. Also, the total volume of foam and liquid was measured 10 min after the operation was stopped and recorded as *V*_10_. Foamability and foam stability were calculated according to Equations (2) and (3), respectively.(2)$$\mathrm{Foaming}\;\mathrm{properties}\;(\%)=\frac{V_{0}-30}{30}\times 100$$(3)$$\mathrm{Foam}\;\mathrm{stability}\;(\%)=\frac{V_{10}-30}{30}\times 100$$
where *V*_0_ represents the foam volume and total liquid volume after 0 min of homogenization, in mL; and *V*_10_ represents the total volume of foam and liquid after 10 min of homogenization, in mL.

#### 2.4.4. Emulsibility and Emulsion Stability Determination

Emulsibility and emulsion stability were determined according to the methods of Xia et al., with modifications [17]. A mixture was prepared by combining 2.0 mL of soybean oil and 8.0 mL of diluted MP solution in a 50 mL centrifuge tube. Following homogenization at 12,000 rpm, a 50 μL aliquot was collected. Then, the aliquot was combined with 5.0 mL of 0.1% (*w*/*v*) SDS solution and analyzed at 500 nm (*A*_0_). After 10 min from being placed in the centrifuge tube, 50 μL of the emulsion was taken, added to 5.0 mL of 0.1% SDS solution and then mixed to determine its absorbance (*A*_10_). The emulsifying activity index (EAI) and emulsifying stability index (ESI) were evaluated based on Equations (4) and (5), respectively.(4)$$\mathrm{EAI}\;(m^{2}/g)=\frac{2\;\times \;2.303\;\times \;A_{0}\;\times \;n}{\rho \times (1-\varphi )\times 10^{4}}$$(5)$$\mathrm{ESI}\;(\%)=\frac{A_{10}}{A_{0}}\times 100$$
where *ρ* represents the mass concentration of MP in the emulsion, in mg/mL; *A*_0_ represents the absorbance of the emulsion at 0 min; *A*_10_ represents the absorbance of the emulsion at 10 min; *φ* represents the oil phase volume fraction (*φ* = 0.2); and *n* represents the dilution ratio.

### 2.5. Observation of MP Microstructure

The microstructure of MP was examined based on the method of Lang et al. [18]. SEM imaging was performed at an acceleration voltage of 20 kV using current Mode 2. The system automatically executed a high-voltage application followed by sequential auto-focusing, brightness adjustment, and contrast optimization through integrated software control. Microstructural observations were conducted at a standardized magnification of 100×.

### 2.6. Methods for Determining the Protein Integrity of MP

#### 2.6.1. SDS-PAGE Analysis

The SDS-PAGE analysis was carried out based on the approach of Huang et al. [19]. Specifically, 20 μL of 5× loading buffer was mixed with 80 μL each of unmodified MP and OSA-modified MP solutions (2 mg/mL). The mixed solutions were heated in a boiling water. Then, a 15 μL aliquot was loaded with the treated protein solution. The resolving gel contained 12% acrylamide, and the stacking gel 5% acrylamide. The PageRuler Prestained Protein Ladder (molecular weight range: 10–250 kDa) served as the molecular weight marker [20]. The first stage was operated in constant voltage mode at 80 V, while the second stage was operated at 120 V. The gel obtained was dyed using Coomassie Brilliant Blue R-250 for about 2 h, destained using a destaining solution until the protein bands were visible and finally photographed with a G BOX F3 GeneSys system (Syngene, UK).

#### 2.6.2. Ca^2+^-ATPase Activity

Exactly 2.00 g of crayfish meat and 9 times the volume of normal saline (approximately, 18 mL) were mixed and homogenized. The supernatant of the centrifugate was diluted to an appropriate concentration with normal saline. Ca^2+^-ATPase activity was measured using a microplate assay kit (ultramicro Ca^2+^-ATPase kit). The specific Ca^2^⁺-ATPase activity of MP was calculated using Equation (6).(6)$${\mathrm{Ca}}^{2+}\text{-}\mathrm{ATPase}\;\mathrm{activity}\;(U/\mathrm{mg})=\frac{(A-A_{0})\;\times \;C_{0}\;\times \;6\;\times \;2.8}{A_{1}\;\times \;A_{2}\;\times \;C}$$
where *A* represents the absorbance values of the test group; *A*_2_, *A*_1_ and *A*_0_ refer to the absorbance values of the blank, standard and control, respectively; and *C*_0_ and *C* represent the concentration of the standard product and the concentration of crayfish MP (mg/mL), respectively.

### 2.7. Determination of MP Conformation

#### 2.7.1. Circular Dichroism Spectrum

The secondary structure was analyzed following the method of Zhang et al. [21]. The prepared MP solutions (0.4 mg/mL) were transferred to a 0.1 cm pathlength quartz cuvette and analyzed using a Jasco J-715 spectropolarimeter (Jasco Corporation, Tokyo, Japan) under the following conditions: 0.25 s response time, 190–260 nm wavelength range, and 100 nm/min scanning speed. Three consecutive scans were averaged to obtain the final spectra.

#### 2.7.2. Contents of Total Thiol and Disulfide Bonds

The total thiol and disulfide bond contents were assessed following the method of Beveridge et al. [22].

Total Thiol A solution: weighed urea, EDTA, SDS and Tris. The solution was standardized to pH 7.0 through titration with hydrochloric acid, followed by volumetric standardization to 1.0 L using distilled water.

Total Thiol B solution: weighed DTNB and Tris. The solution was standardized to pH 8.0 through titration with hydrochloric acid, followed by volumetric standardization to 1.0 L using distilled water.

Measurement: The MP solution (1.0 mL) was diluted with B solution. The total thiol A solution (9.0 mL) was added, and the mixture was stirred. The total thiol B solution (0.4 mL) was added, and the mixture was incubated in a water bath (40 °C for 25 min). The absorbance of the solution was recorded at 412 nm with 0.2 M PBS buffer as the reference blank control. The total thiol content of MP was derived from Equation (7).(7)$$\mathrm{Total}\;\mathrm{thiol}\;\mathrm{molar}\;\mathrm{concentration}\;(\mathrm{mol}/g)=\frac{A\;\times \;D}{\varepsilon \;\times \;C_{0}}$$
where *C*_0_ represents MP concentration, in mg/mL; *A* represents the absorption value at 412 nm; *D* represents the dilution ratio; and *ε* represents the molar extinction coefficient, 13,600 L/(mol·cm).

Disulfide bond A solution: weighed urea, EDTA, SDS and K_2_SO_3_. The solution was standardized to pH 8.0 through titration with hydrochloric acid, followed by volumetric standardization to 1.0 L using distilled water.

Disulfide bond B solution: weighed DTNB and K_2_SO_3_. The solution was standardized to pH 9.5 through titration with hydrochloric acid, followed by volumetric standardization to 1.0 L using distilled water.

The disulfide bond content was measured using the same methodology as that for total thiol determination, with modifications to the solution compositions, as specified above.

#### 2.7.3. Determination of Surface Hydrophobicity

The surface hydrophobicity was determined according to the methods of Lv et al. [23]. For the experimental groups, 500 μL of a 1 mg/mL MP solution and 100 μL of a bromophenol blue (BPB) solution (1 mg/mL) were mixed. For the blank controls, 500 μL of 0.6 M KCl-20 mmol/L Tris-maleic acid buffer (pH 7.0) and 100 μL of BPB solution (1 mg/mL) were mixed. The mixtures were centrifuged subsequently, and 150 μL of the supernatants were diluted to 1.5 mL. The absorbance was measured at 595 nm using a UV spectrophotometer. The surface hydrophobicity of MP was calculated using Equation (8).(8)$$\mathrm{Surface}\;\mathrm{hydrophobicity}\;(\mu g/\mathrm{mg})=\frac{200\;\times \;(A_{1}-A_{2})}{A_{1}}$$
where *A*_1_ represents the blank absorbance; and *A*_2_ represents the absorbance of MP.

#### 2.7.4. Determination of Fluorescence Intensity

Fluorescence intensity was determined following the method of Chung et al. [24]. Weigh 2.4228 g of Tris and 44.73 g of KCl, dissolve them in 800 mL of distilled water while stirring, adjust the pH to 7.0 with hydrochloric acid, transfer the solution to a 1000 mL volumetric flask and dilute it to the mark with distilled water to prepare the extraction solution. The extraction solution served as the blank control, and fluorescence intensity was determined with an F-7000 fluorometer. Excitation wavelength, slit width and emission wavelength were set at 295 nm, 2.5 nm and 300–500 nm, respectively.

### 2.8. Data Analysis

All experiments were performed in triplicate, and the data are reported as mean ± standard deviation (SD). Statistical analysis was performed using SPSS 19.0 (IBM, New York, NY, USA) and Origin 2022 (OriginLab, Northampton, MA, USA). Statistical evaluations were conducted through one-factor ANOVA complemented by Duncan’s post hoc comparisons (SPSS 25.0, IBM), where *p* < 0.05 was considered statistically significant.

## 3. Results

### 3.1. Functional Properties of MP

#### 3.1.1. Solubility

The solubility of MP is one of the key indicators reflecting oxidation and deterioration in crayfish, as it directly affects the stability, emulsification and gelation properties of muscle tissues, thereby influencing their structural properties [25]. The solubility of MP was significantly decreased (*p* < 0.05) following the FT cycles (Figure 1A). The turbidity values showed a progressive increase with the number of FT cycles (Figure 1B). Compared to MP from the other treatments, MP treated with LNF consistently exhibited the highest solubility during the FT cycles. The solubility of LNF-treated MP remained at 57.1%, while that of MP from the RF groups (−20 and −50 °C) exhibited values below 45% after five FT cycles. Notably, MP solubility following RF was significantly lower (*p* < 0.05) than following LNF after three FT cycles. This decrease in solubility may have resulted from protein denaturation or aggregation during the FT cycles. The lower freezing temperature (LNF) induced the formation of smaller ice crystals, which minimized the damage to the muscle cells and MP [26]. The results indicated that repeated FT cycles exacerbate MP denaturation, which might change the protein microstructure and induce protein oxidation [27]. Pork MP solubility dropped by about 50% after five FT cycles [17]. However, LNF-treated crayfish MP solubility remained at 57.1% after five FT cycles. After three FT cycles, the solubility of cod MP dropped to 65%, compared to 68% for crayfish MP under the same conditions [28]. This indicates that the difference in freeze resistance between crustacean and fish proteins may be determined by muscle fiber structure. LNF may mitigate the structural denaturation of MP during repeated FT cycles.

#### 3.1.2. Foaming Properties

The foaming properties are related to the molecular weight, conformation and solubility of proteins [29]. The results of the foaming properties analysis are depicted in Figure 1C,D. The results indicated that the foaming properties and foam stability of MP were reduced during the FT cycles. After five FT cycles, the foaming properties and foam stability of MP treated at −20 °C RF had significantly decreased (*p* < 0.05) by 37.1% and 30.7%, respectively. MP treated at −50 °C RF and −80 °C LNF maintained higher foaming properties than MP subjected to −20 °C RF after five FT cycles. Research has shown that hydrophobic and thiol groups could be exposed during freezing, leading to the formation of non-covalent aggregates, which ultimately impair protein interfacial properties and molecular diffusion rates [30]. Similar findings were reported for tilapia MP, for which foaming capacity progressively declined during storage due to progressive protein denaturation [31]. The initial foaming capacity of egg white protein (120%) was significantly higher than that of crayfish MP (85%) [29]. However, after an FT cycle, the foaming properties of egg white protein decreased more rapidly, with a 60% loss after five FT cycles. In contrast, those of LNF-treated crayfish MP only decreased by 30%. These results suggest that LNF may mitigate MP denaturation, thereby preserving its foaming capacity and foam stability during FT cycles.

#### 3.1.3. Emulsification Properties

Protein emulsification activity and stability depend on protein solubility and extent of denaturation [32]. The results of the EAI and ESI are displayed in Figure 1E,F. The EAI and ESI of MP were significantly decreased (*p* < 0.05) following the FT cycles. Compared to NF MP, after five FT cycles, the EAI and ESI of MP treated at −20 °C RF showed significant reductions (*p* < 0.05) of 61.8% and 51.4%, respectively. Although the EAI and ESI of MP subjected to LNF were also decreased after five FT cycles, the emulsification properties remained superior to those of RF-treated MP. The lower freezing temperature of LNF contributed to maintaining the solubility of MP (Figure 1A), which enhanced its emulsification performance. Studies indicated that freezing storage compromises MP structural integrity, exposing internal MP molecules and reducing MP emulsifying capacity [33]. Furthermore, the FT process induces the formation of intermolecular disulfide bonds and oxidation reactions, which also destroy the emulsifying capacity of MP [34]. Kieserling et al. [31] found that the ESI of whey protein was decreased by 40% after a FT cycle. In comparison, the ESI of crayfish MP only decreased by 25%, indicating a unique interfacial activity advantage for crayfish proteins. The structural disruption of MP likely hinders the formation of stable interfacial membranes. Thus, LNF may preserve crayfish MP functional properties from the alterations in disulfide bonds, hydrogen bonds and hydrophobic interactions. In addition, when using LNF, crayfish MP is likely better suited as a natural emulsifier or foaming agent in frozen foods, especially in products like pre-packaged soups and frozen pastries that require multiple FT cycles.

### 3.2. Microstructural Analysis of MP

The results of the microstructural analysis are depicted in Figure 2. NF crayfish MP showed a smooth and compact surface. The degree of surface wrinkling progressively intensified with the increasing number of FT cycles, which was accompanied by a continuous enlargement of the surface pores. Compared to the −20 °C and −50 °C RF groups, the microstructure of LNF crayfish MP showed a more compact surface and smaller pores. This might have been the result of the formation of smaller ice crystals in the muscle tissue following LNF, which could reduce the damage to MP structure [35]. During the five FT cycles, the particle size, distribution, orientation, and shape of the ice crystals in crayfish muscle continuously changed, leading to protein aggregation, cross-linking, molecular rearrangement and the irreversible denaturation of MP. The recrystallization of ice crystals during the FT cycles could induce a mechanical force on MP, resulting in discontinuous tissue organization, structural loosening and the development of irregular surface pores. This mechanism aligns with observations in carp muscle, where FT cycles caused both intracellular and extracellular ice crystal formation, subsequently disrupting the native tissue microstructure [32]. Notably, LNF significantly mitigated the formation of internal surface pores and surface wrinkling compared to conventional RF.

### 3.3. Molecular Weight of MP and Ca^2+^-ATPase Activity Analysis

#### 3.3.1. SDS-PAGE

The FT cycles induce the degradation or aggregation of protein molecules, which is typically manifested as the disappearance or intensification of protein bands in SDS-PAGE [36]. MP contains myosin heavy chain, myosin light chain, actin, and tropomyosin [37]. The results of the molecular weight distribution analysis are shown in Figure 3. With prolonged frozen storage and an increasing number of FT cycles, the protein bands became progressively narrower and fainter. Similar studies on white shrimp demonstrated that progressive FT cycles promote the oxidation and degradation of MP [38]. Compared to LNF and −50 °C RF, MP treated at −20 °C RF exhibited the most attenuated band intensity throughout the FT cycles. Notably, the average molecular weight of LNF-treated MP retained the highest values, suggesting reduced degradation.

#### 3.3.2. Ca^2+^-ATPase Activity

Ca^2+^-ATPase activity is a key indicator to assess the extent of protein denaturation [39]. The myosin head exhibits sensitivity to FT cycles, which results in a reduction in Ca^2^⁺-ATPase activity. The effects of the freezing methods and FT cycles on Ca^2+^-ATPase activity are presented in Figure 4. Compared to the NF group, no significant decrease (*p* > 0.05) in Ca^2^⁺-ATPase activity was observed in frozen crayfish after three FT cycles. Beyond this point, all groups exhibited significant declines (*p* < 0.05) in activity with the increase in the FT cycles. Lower freezing temperatures attenuated the reduction rate of Ca^2^⁺-ATPase activity. Progressive ice crystal growth and recrystallization during freezing could induce conformational changes in myosin heads, which directly correlated with diminished Ca^2^⁺-ATPase activity [40]. Additionally, low-temperature-induced molecular reorganization and thiol oxidation further contribute to this decline [41]. Consistent with these studies, sulfhydryl group denaturation and oxidation in myosin heads could significantly impair Ca^2^⁺-ATPase activity during crayfish freezing [42]. Notably, LNF could preserve myosin head conformation and mitigate aggregation during the FT cycles, maintaining crayfish quality.

### 3.4. MP Conformation Analysis

#### 3.4.1. Secondary Structure

The content of *α*-helices could be used to judge the stability of a protein structure [43]. The orderliness of the MP molecule transitioned toward disorder following the FT cycles (Table 1). Compared to MP under −20 °C and −50 °C RF, more random coils were generated in LNF-pretreated crayfish MP. The disruption of hydrogen bonds between carbonyl oxygen and amino oxygen was attributed to the generation of larger ice crystals, leading to the irreversible denaturation of the MP [44]. Studies indicate that an increase in protein oxidation could lead to a decrease in the content of *α*-helices [34]. After three FT cycles, the *α*-helix content in LNF crayfish MP (15.00% ± 1.85%) was significantly higher than that in MP after −20 °C RF (9.32% ± 0.89%) and −50 °C RF (10.18% ± 1.41%). This implies that LNF could effectively protect crayfish MP from oxidation during the FT cycles.

#### 3.4.2. Total Sulfhydryl Group and Disulfide Bond Contents

Cysteine and methionine contain abundant thiol groups that are prone to oxidative modification. Oxidized sulfhydryl groups may undergo intermolecular reactions to form disulfide bonds or interact with oxygen to generate thiosulfinate radicals [45]. As showed in Figure 5, the total sulfhydryl (T-SH) content in MP was significantly decreased (*p* < 0.05), while the disulfide bond content was significantly increased (*p* < 0.05) following the FT cycles. The decline in total thiols might have resulted from the mechanical damage induced by ice crystal particles, exposing the internal thiol groups and oxidizing them into disulfide bonds [46,47]. Compared to the control (NF MP), the total thiol content in MP subjected to −20 °C RF, −50 °C RF and LNF was reduced to 29.8, 36.5 and 40.8 mol/10^5^ g, and the disulfide bond content was increased to 35.7, 32.5 and 28.5 mol/10^5^ g, respectively. These results suggest that LNF reduced the exposure of the internal sulfhydryl groups, thereby mitigating MP denaturation. A similar result was reported for black pork MP, indicating that the total thiol content tended to decrease with the increase in the number of FT cycles, which was caused by the recrystallization of ice crystals [26]. The reduction in sulfhydryl content may have resulted from the generation of S-S bonds in the sulfhydryl groups. [48]. The accelerated denaturation of myosin, particularly its conformational changes, may facilitate the formation of disulfide bonds [49]. The results of Ca^2+^-ATPase activity analysis also showed that LNF could lessen the extent of myosin conformational changes. Compared to RF, LNF inhibited the formation of large ice crystals and prevented the destruction of MP spatial structure, thus reducing the exposure of T-SH and indirectly inhibiting the conversion of thiol to disulfide.

#### 3.4.3. Surface Hydrophobicity

Protein surface hydrophobicity serves as an indicator of the extent to which hydrophobic groups are exposed. [50]. As displayed in Figure 5A, the surface hydrophobicity of MP increased significantly (*p* < 0.05) as the FT cycles progressively increased. Notably, RF crayfish MP exhibited more pronounced increases in surface hydrophobicity compared to LNF MP. After five FT cycles, the surface hydrophobicity of −20 °C RF crayfish MP increased 2.75 times compared to that of NF MP (76.55 μg/mg). Freezing-induced denaturation of MP during frozen storage might expose the internal hydrophobic groups [51]. FT cycles disrupted the structure of cured pork MP and exposed the hydrophobic groups, causing an enhancement in surface hydrophobicity. These results suggest that LNF could reduce the denaturation of MP during frozen storage and prevent MP from undergoing excessive folding and aggregation.

#### 3.4.4. Fluorescence Intensity

Tryptophan (Trp) fluorescence characteristics are utilized to detect alterations in protein tertiary structure. The results of the fluorescence intensity analysis of crayfish MP are presented in Figure 6B–D. The fluorescence intensity of MP decreased following the FT cycles, which indicated a change in MP structure. The protein structure unfolded, the internal hydrophobic environment changed, and Trp residues were exposed, leading to a decrease in fluorescence intensity [52]. The variation trend of fluorescence intensity was in line with that of surface hydrophobicity, indicating that the hydrophobic groups were increasingly exposed with the increase in freeze–thaw cycles. It can be seen that LNF MP maintained a higher fluorescence intensity than MP after −20 °C and −50 °C RF during the FT cycles. The decrease in fluorescence intensity of MP was mainly due to increased steric hindrance, which was caused by the aggregation of MP and the rise in hydrophobic interactions. Changes in Trp residues inside the chains of myosin could cause MP aggregation, leading to the freezing-induced denaturation of MP. The results of the surface hydrophobicity and fluorescence intensity analyses indicated that LNF can delay the degree of denaturation of MP.

## 4. Conclusions

LNF significantly alleviated the changes in the conformation and functional characteristics of MP caused by repeated FT cycles. By forming smaller ice crystals, LNF minimizes the mechanical damage to muscle tissues, thereby ensuring the integrity of proteins. A microstructure analysis further confirmed that LNF prevented excessive surface wrinkling and pore formation in MP. Additionally, LNF effectively delayed the transition of ordered *α*-helix structures to disordered conformations, which highlights its ability to suppress protein denaturation and aggregation. From the perspective of MP conformational changes, LNF maintained MP molecular orderliness, especially by protecting the myosin head and hydrophobic groups, thereby preventing protein aggregation and freezing-induced denaturation. Despite the obvious protective effect on the quality of crayfish, there are still some problems in practical applications. For example, it is not possible to ensure that all sale points have LNF equipment. Therefore, improvements are still needed in the future in terms of the supply of small-scale LNF equipment and the construction of logistics bases. In addition, new technologies for reducing the cost of LNF are also necessary in the future.

## Figures and Tables

**Figure 1 foods-14-02078-f001:**
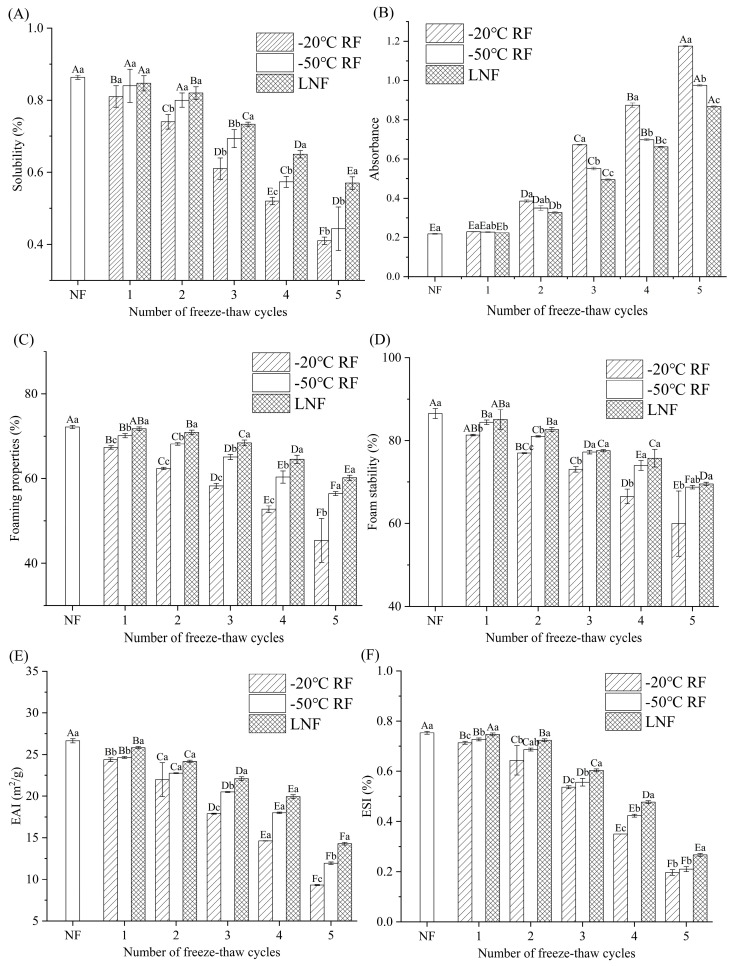
Effect of freezing methods on the solubility (**A**), turbidity (**B**), foaming properties (**C**), foam stability (**D**), emulsifying activity index (**E**) and emulsifying stability index (**F**) of crayfish myofibrillar protein. Distinct letters on the same column signify a significant difference (*p* < 0.05). Lowercase letters and uppercase letters represent statistical significance for the differences between freezing methods and freeze–thaw cycles, respectively. NF indicates non-frozen (
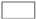
); RF indicates refrigerator freezing, −20 °C RF (
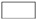
), −50 °C RF (
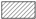
); LNF indicates −80 °C liquid nitrogen freezing (
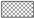
). The numbers 1, 2, 3, 4 and 5 indicate frozen crayfish subjected to 1, 2, 3, 4 and 5 freeze–thaw cycles.

**Figure 2 foods-14-02078-f002:**
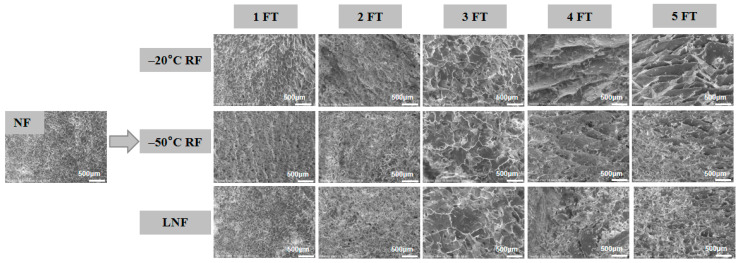
Microstructure of myofibrillar protein gel. NF indicates non-frozen; RF indicates refrigerator freezing, −20 °C RF, −50 °C RF; LNF indicates −80 °C liquid nitrogen freezing. The numbers 1, 2, 3, 4 and 5 indicate frozen crayfish subjected to 1, 2, 3, 4 and 5 freeze–thaw cycles.

**Figure 3 foods-14-02078-f003:**
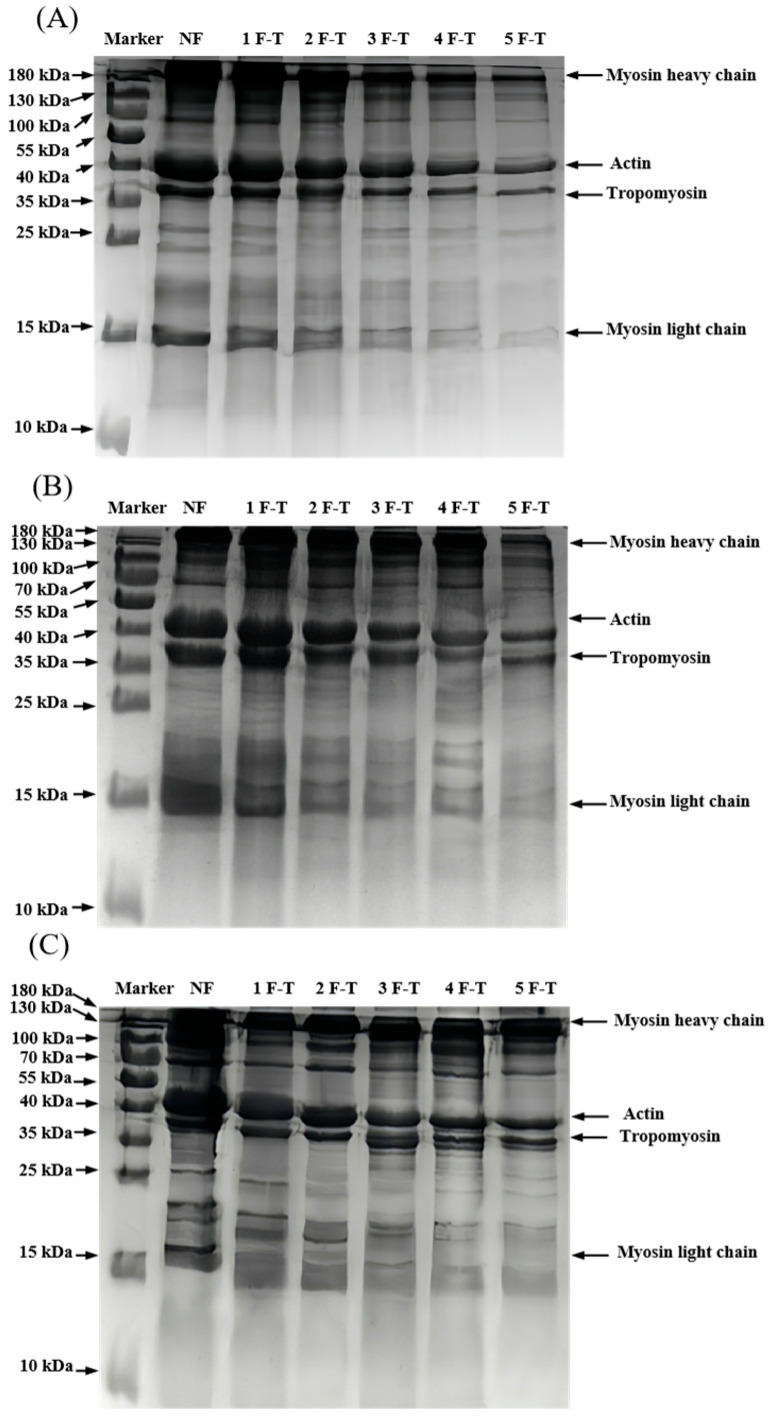
Molecular weight profiles of crayfish myofibrillar protein under −20 °C RF (**A**), −50 °C RF (**B**) and −80 °C LNF (**C**). NF indicates non-frozen; RF indicates refrigerator freezing, −20 °C RF, −50 °C RF; LNF indicates −80 °C liquid nitrogen freezing. The numbers 1, 2, 3, 4 and 5 indicate frozen crayfish after 1, 2, 3, 4 and 5 freeze–thaw cycles.

**Figure 4 foods-14-02078-f004:**
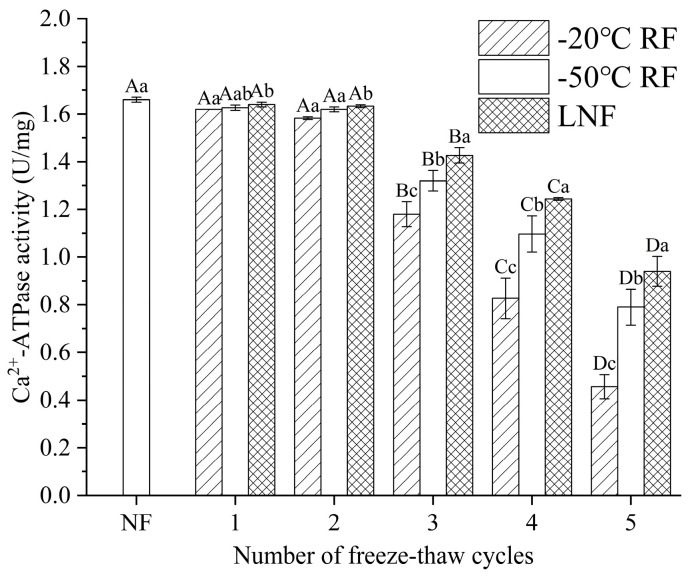
Ca^2+^-ATPase activity of crayfish myofibrillar protein with different freezing methods. Distinct letters in the same column signify a significant difference (*p* < 0.05). Lowercase letters and uppercase letters represent statistically significant differences between freezing methods and freeze–thaw cycles, respectively. NF indicates non-frozen (
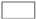
); RF indicates refrigerator freezing, −20 °C RF (
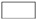
), −50 °C RF (
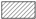
); LNF indicates −80 °C liquid nitrogen freezing (
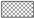
). The numbers 1, 2, 3, 4 and 5 indicate frozen crayfish after 1, 2, 3, 4 and 5 freeze–thaw cycles.

**Figure 5 foods-14-02078-f005:**
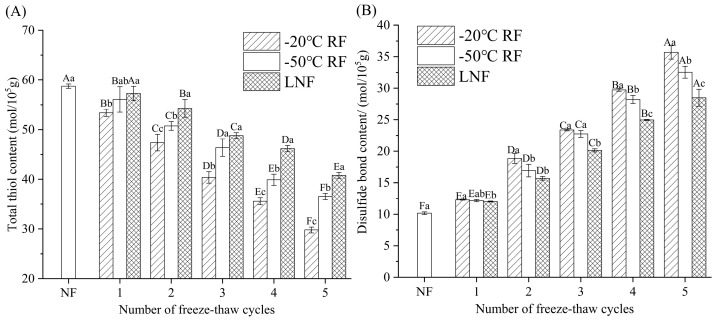
Total thiol (**A**) and disulfide bond (**B**) contents in crayfish myofibrillar protein under different freezing conditions. Distinct letters in the same column signify a significant difference (*p* < 0.05). Lowercase letters and uppercase letters represent statistically significant differences between freezing methods and freeze–thaw cycles, respectively. NF indicates non-frozen crayfish (
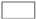
); RF indicates refrigerator freezing, −20 °C RF (
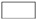
), −50 °C RF (
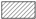
); LNF indicates −80 °C liquid nitrogen freezing (
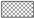
). The numbers 1, 2, 3, 4 and 5 indicate frozen crayfish subjected to 1, 2, 3, 4 and 5 freeze–thaw cycles.

**Figure 6 foods-14-02078-f006:**
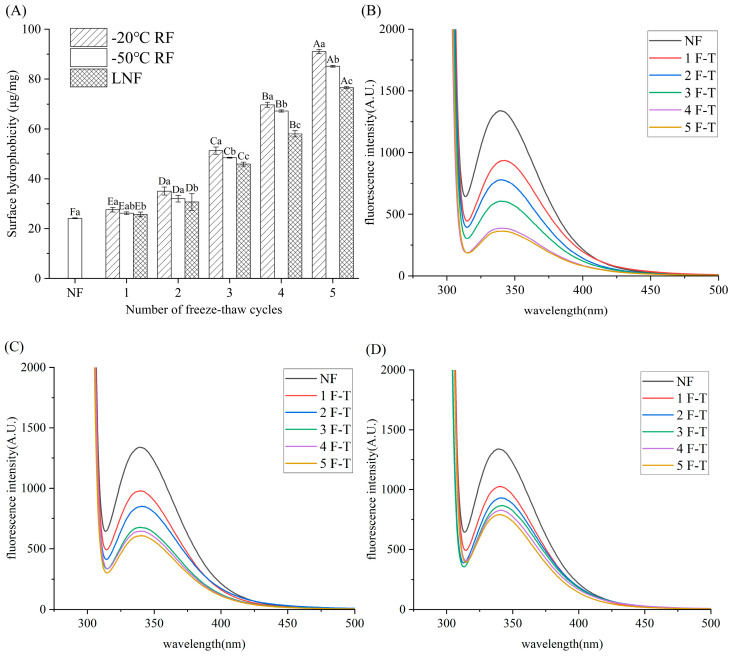
Surface hydrophobicity (**A**) and fluorescence intensity of crayfish myofibrillar protein frozen at −20 °C RF (**B**), −50 °C RF (**C**) and LNF (**D**). Distinct letters in the same column signify a significant difference (*p* < 0.05). Lowercase letters and uppercase letters represent statistically significant differences between freezing methods and freeze–thaw cycles, respectively. NF indicates non-frozen crayfish (
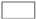
); RF indicates refrigerator freezing, −20 °C RF (
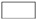
), −50 °C RF (
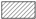
); LNF indicates −80 °C liquid nitrogen freezing (
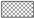
). The numbers 1, 2, 3, 4 and 5 indicate frozen crayfish after 1, 2, 3, 4 and 5 freeze–thaw cycles.

**Table 1 foods-14-02078-t001:** The secondary structure of myofibrillar protein subjected to different freezing methods and a different number of freeze–thaw cycles.

Crayfish		Secondary Structure			
Temperature	Freeze–Thaw	*α*-Helix	*β*-Turn	*β*-Sheet	Random Coil
−20 °C	1	24.32 ± 2.20 ^Aa^	36.27 ± 2.36 ^Aa^	28.51 ± 2.13 ^Cb^	10.91 ± 2.10 ^Ca^
	2	21.05 ± 1.05 ^Aa^	28.80 ± 0.43 ^Bb^	29.62 ± 2.00 ^Cc^	20.54 ± 0.94 ^Ba^
	3	9.32 ± 0.89 ^Bb^	18.86 ± 0.69 ^Cb^	47.08 ± 1.18 ^Bb^	24.74 ± 0.38 ^Aa^
	4	9.47 ± 0.49 ^Ba^	17.97 ± 1.55 ^Ca^	61.09 ± 1.70 ^Aa^	11.47 ± 0.55 ^Cc^
	5	6.00 ± 0.51 ^Cb^	10.58 ± 1.13 ^Db^	63.05 ± 1.08 ^Aa^	20.37 ± 1.16 ^Bb^
−50 °C	1	22.19 ± 2.59 ^Aa^	28.34 ± 1.74 ^Ab^	37.84 ± 1.22 ^Da^	11.64 ± 0.92 ^Ca^
	2	21.19 ± 2.24 ^Aa^	25.53 ± 1.82 ^Bb^	41.52 ± 1.37 ^Ca^	11.44 ± 0.86 ^Cb^
	3	10.18 ± 1.41 ^Bb^	21.17 ± 1.80 ^Cb^	54.42 ± 1.99 ^Ba^	14.24 ± 0.58 ^Bb^
	4	6.72 ± 1.36 ^Cb^	15.67 ± 0.64 ^Cb^	56.04 ± 1.21 ^Bb^	21.57 ± 1.31 ^Ab^
	5	5.43 ± 0.68 ^Cb^	20.88 ± 0.25 ^Da^	59.40 ± 0.61 ^Ab^	14.28 ± 0.74 ^Bc^
−80 °C	1	23.36 ± 2.03 ^Aa^	36.15 ± 1.13 ^Aa^	28.44 ± 1.55 ^Db^	11.05 ± 1.16 ^Ca^
	2	22.75 ± 0.71 ^Aa^	34.97 ± 2.67 ^Aa^	33.20 ± 1.65 ^Cb^	9.08 ± 1.24 ^Cc^
	3	15.00 ± 1.85 ^Ba^	35.00 ± 2.32 ^Aa^	35.00 ± 2.15 ^Cc^	15.00 ± 1.24 ^Bb^
	4	9.00 ± 0.74 ^Ca^	19.00 ± 0.41 ^Ba^	47.00 ± 1.28 ^Bc^	25.00 ± 1.13 ^Aa^
	5	7.93 ± 0.10 ^Ca^	11.55 ± 0.49 ^Cb^	54.63 ± 0.57 ^Ac^	26.06 ± 2.51 ^Aa^

Note: in the same column, different lowercase letters indicate significant differences among freezing methods, and different uppercase letters indicate significant differences among freeze–thaw cycles, as determined by Duncan’s multiple range test (*p* < 0.05). Data are presented as mean ± standard deviation.

## Data Availability

The original contributions presented in the study are included in the article; further inquiries can be directed to the corresponding authors.

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
