# Peer review of "Understanding the Protective Effect of Liquid Nitrogen Freezing on Crayfish Quality During Transportation and Storage"

_foods, 2025, doi:10.3390/foods14122078_

Round 1
Reviewer 1 Report
Comments and Suggestions for Authors
This is an interesting, extensive, and informative work. Below are comments or observations that the authors must address in the document.
1. The authors are encouraged to improve their conclusions by explaining the limitations of the work and future research perspectives derived from the study.
2. In sections 3.4.4. Fluorescence intensity and 3.4.2. Total thiol and disulfide bond contents, the figures cited do not correspond. Review the entire document to ensure the cited figures match.
3. Include the web access link in the references section.
4. What would be the practical application of the information obtained from the study or recommendations for the food industry?
5. Include in the introduction or conclusions: Why, despite the knowledge of the use of Liquid Nitrogen Freezing (LNF) for food preservation, has it not been regularly implemented for this type of product? Are there any disadvantages to using this method? 6. It is suggested that a comprehensive experimental diagram of the study be developed and included in the materials and methods section.
7. What were the characteristics of the crayfish used in the study (weight, length, etc.)?
7. Was the extracted protein content (MP) quantified? Yes, no, why?
8. In the functional properties results section, the authors are encouraged to compare their results with studies conducted on other types of animal proteins.
Comments on the Quality of English Language
The manuscript must have an english language review.
Author Response
Comments 1: The authors are encouraged to improve their conclusions by explaining the limitations of the work and future research perspectives derived from the study.
Response 1: Thank you for your valuable comments. We revised the conclusion partly.
Lines 443-447: Despite the obvious protective effect on the quality of crayfish, there are still some problems. For example, it cannot ensure that all sales links have LNF equipment. Therefore, improvements are still needed in the future in terms of small-scale LNF equipment and the construction of logistics bases. In addition, the research may focus on the new technologies for reducing the cost of LNF.
Comments 2: In sections 3.4.4. Fluorescence intensity and 3.4.2. Total thiol and disulfide bond contents, the figures cited do not correspond. Review the entire document to ensure the cited figures match.
Response 2: Thank you for your valuable comments. We have revised them.
Line 372: As showed in Figure 5, the total sulfhydryl (T-SH) of MP was significantly decreased (p < 0.05).
Line 411: The results of fluorescence intensity of crayfish MP was presented in Figure 6B-D.
Comments 3: Include the web access link in the references section.
Response 3: Thanks for your valuable comments. We added web access link with every reference.
Comments 4: What would be the practical application of the information obtained from the study or recommendations for the food industry?
Response 4: Liquid nitrogen freezing (LNF) can significantly reduce the mechanical damage of ice crystals to myofibrillar proteins (MPs) from aquatic products (such as crayfish, fish, etc.), thereby maintaining the texture, taste and nutritional value, and providing a scientific solution for the frozen preservation of aquatic products during transportation and storage.
Comments 5: Include in the introduction or conclusions: Why, despite the knowledge of the use of Liquid Nitrogen Freezing (LNF) for food preservation, has it not been regularly implemented for this type of product? Are there any disadvantages to using this method? It is suggested that a comprehensive experimental diagram of the study be developed and included in the materials and methods section.
Response 5: Thanks for your valuable comments. For the requirement of the number of pictures, we added discussion about the disadvantages of LNF.
Lines 35-41: It has advantages with excellent preservation effect, long shelf-life extension, low dry consumption, small footprint, wide applicability and environmental friendliness for food preservation. However, it also has disadvantages such as high cost, high equipment requirements and high storage and transportation requirements, which limited the application of LNF for low-value products. For some high-value products, such as crayfish, more rapid freezing speed contributes to preventing the formation of large ice crystals, which is advantage for maintaining crayfish quality.
lines 443-447: Despite the obvious protective effect on the quality of crayfish, there are still some problems. For example, it cannot ensure that all sales links have LNF equipment. Therefore, improvements are still needed in the future in terms of small-scale LNF equipment and the construction of logistics bases. In addition, the research may focus on the new technologies for reducing the cost of LNF.
Comments 6: What were the characteristics of the crayfish used in the study (weight, length, etc.)?
Response 6: We added the characteristics and weight of the crayfish samples used.
Line 79: Live crayfish (similar in size, bright red in color, and weighing 20.0 ± 5.00 g) were frozen in foam boxes with ice packs, and transported to the laboratory within 2 h.
Comments 7: Was the extracted protein content (MP) quantified? Yes, no, why?
Response 7: Yes, the extracted MP content was quantified. Biuret method was used to measure total protein concentration of extracted MP. Different quantified protein were used for every experiments, such as 1 mg/mL for solubility, 2 mg/mL for SDS-PAGE.
Comments 8: In the functional properties results section, the authors are encouraged to compare their results with studies conducted on other types of animal proteins.
Response 8: Thanks for your comments. We have included the comparison in section 3.1.
Lines 238-242: Pork MP solubility drops by about 50% after 5 FT cycles[17]. However, LNF-treated crayfish MP solubility remains at 57.1% after 5 FT cycles. After 3 FT cycles, the solubility of cod MP drops to 65%, compared to 68% for crayfish MP with the same conditions[35].
Lines 256-259: The initial foaming capacity of egg white protein (120%) was significantly higher than that of crayfish MP (85%)[28]. However, after FT cycle, the foaming properties of egg white protein were decreased more rapidly, with a 60% loss after 5 FT cycles. In contrast, the LNF-treated crayfish MP only decreased by 30%.
Lines 275-278: Kieserling et al.[31] found that the ESI of whey protein was decreased by 40% after FT cycle. In comparison, the ESI of the crayfish MP only was decreased by 25%, indicating a unique interfacial activity advantage of crayfish proteins.
Lines 280-283: In addition, crayfish MP is likely better suited as a natural emulsifier or foaming agent with frozen foods, especially pre-packaged soups and frozen pastries that require multiple FT cycles.
Reference
[17] Xia X., Kong B., Xiong Y., Ren Y. Decreased gelling and emulsifying properties of myofibrillar protein from repeatedly frozen-thawed porcine longissimus muscle are due to protein denaturation and susceptibility to aggregation. Meat Science 2010, 85, 481-486.
[35] Gao W., Hou R., Zeng X. Synergistic effects of ultrasound and soluble soybean polysaccharide on frozen surimi from grass carp. Journal of Food Engineering 2019, 240, 1-8.
[28] Gharbi N., Labbafi M. Influence of treatment-induced modification of egg white proteins on foaming properties. Food Hydrocolloids 2019, 90, 72-81.
[31] Kieserling H., Pankow A., Keppler J. K., Wagemans A. M., Drusch S. Conformational state and charge determine the interfacial film formation and film stability of β-lactoglobulin. Food Hydrocolloids 2021, 114, 106561.

Reviewer 2 Report
Comments and Suggestions for Authors
I am writing to submit my review of the manuscript entitled "Understanding the protective mechanism of liquid nitrogen freezing on the crayfish quality during transportation and storage" for your consideration. Overall, I find the manuscript's findings intriguing, and the information provided helpful for researchers and academia. The article has the potential to make a significant contribution to the related discipline. However, some comments should be addressed.
Line 45-46: The statement about temperature fluctuations during transportation and storage is vague. Provide specific temperature ranges or conditions to clarify the extent of fluctuations.
Line 52-53: The claim that mechanical damage is the "most important factor" for MP denaturation is strong. Cite comparative studies or quantify its contribution relative to other factors (e.g., oxidation).
Line 64-67: The sources of crayfish and liquid nitrogen are mentioned, but details like size, weight, or handling conditions of crayfish are missing. Standardize sample descriptions for reproducibility.
Line 75-81: The freezing protocols (-20°C, -50°C, -80°C) lack justification. Explain why these temperatures were chosen and how they compare to industry standards.
Line 104-113: Foaming properties are described, but the homogenization speed/duration is omitted. Specify these to ensure consistency in foam formation.
Line 137-145: SDS-PAGE methodology lacks gel concentration (% acrylamide) and marker details. Include these for clarity and reproducibility.
Line 211-226: Solubility results are discussed, but statistical significance markers (e.g., p-values) are missing in Figure 1A. Add these to support claims.
Line 256-264: Figure 1 captions should explicitly state the number of replicates (n) for each data point to assess reliability.
Line 340-355: Table 1 notes "Distinct letters" for significance but does not explain the statistical test used (e.g., Duncan’s, Tukey’s). Clarify in the caption.
Line 363-377: Surface hydrophobicity results (Figure 6A) should correlate with fluorescence data (Figure 6B-D). Discuss this relationship explicitly.
Line 404-413: The conclusion states LNF "significantly alleviated" MP changes but does not quantify "significant." Define effect sizes or percent improvements.
Line 414-420: The funding statement mentions a grant but does not link it to specific experiments or outcomes. Clarify the grant’s role in the study.
General comments:
Abstract: The abstract should succinctly capture the essence of the research. Consider focusing on key findings and implications for clarity.
Language and readability: The manuscript's language is often difficult to follow. Several sections lack critical information, making it challenging for readers to grasp the study's findings and implications. A thorough language edit is recommended to improve clarity and readability.
Conclusion: Improve the conclusion by focusing on the main findings and writing future remarks and notes for commercial applications.
Overall: The manuscript needs massive English corrections
Line 45-46: "temperature fluctuation would inevitably occur" → "temperature fluctuations inevitably occur" (simplify tense).
Line 104-105: "Foaming properties and foaming stability were analyzed following the method of Sharifian et al.[16] with modifications" → Specify modifications for clarity.
Line 152-154: The equation’s description is fragmented. Revise for fluency: "where A, A₀, A₁, and A₂ represent the absorbance values of..."
Line 256-264: Figure 1 caption uses inconsistent formatting (e.g., "n FT"). Standardize to "nFT" or "n-FT."
Technical Terms
Line 52: "cytoclasis" is uncommon. Replace with "cell rupture" or "lysis."
Line 363: "T-SH" is undefined. Spell out "total sulfhydryl groups" at first use.
Avoid passive voice where active is clearer (e.g., "It was reported that..." → "Studies indicate...").
Define abbreviations (e.g., "MP") at first use in the abstract and main text.
You may highlight the economic importance of LNF and provide recommendations for implementation.
Notably, by optimizing LN₂ use, hybridizing systems, and leveraging economies of scale, food producers can make LNF cost-competitive. The key is balancing quality preservation with strategic cost-cutting measures, especially for high-value products like seafood, berries, and premium meats
Comments on the Quality of English Language
English needs massive correction.
Author Response
Comments 1: Line 45-46: The statement about temperature fluctuations during transportation and storage is vague. Provide specific temperature ranges or conditions to clarify the extent of fluctuations.
Response 1: Thanks for your valuable comment. We consulted relevant literature and determined the temperature fluctuation range.
Line 50: Nevertheless, due to the time required to market these frozen crayfish, temperature fluctuations (-18 °C to -5 °C) inevitably occur during transportation and storage.
Reference: [7] Zhang Y., Bai G., Wang J., Wang Y., Jin G., Teng W., Geng F., Cao J. Myofibrillar protein denaturation/oxidation in freezing-thawing impair the heat-induced gelation: Mechanisms and control technologies. Trends in Food Science & Technology 2023, 138, 655-670.
Comments 2: Line 52-53: The claim that mechanical damage is the "most important factor" for MP denaturation is strong. Cite comparative studies or quantify its contribution relative to other factors (e.g., oxidation).
Response 2: Thanks for your valuable comments. We revised the statement.
Lines 56-59: Formation of crayfish quality degradation generally caused by damage of muscle fibers resulting from ice crystallization and the freezing-induced denaturation of MP, which further affects the color, flavor and taste.
Comments 3: Line 64-67: The sources of crayfish and liquid nitrogen are mentioned, but details like size, weight, or handling conditions of crayfish are missing. Standardize sample descriptions for reproducibility.
Response 3: Thanks for your valuable comments. We added the details of crayfish samples.
Lines 82-83: Live crayfish (similar in size, bright red in color, and weighing 20.0±5.00 g) were frozen in foam boxes with ice packs, and transported to the laboratory within 2 h.
Comments 4: Line 75-81: The freezing protocols (-20°C, -50°C, -80°C) lack justification. Explain why these temperatures were chosen and how they compare to industry standards.
Response 4: -20°C is a common temperature for household and some industrial freezing equipment (e.g., cold chain transport) and is widely used for preserving aquatic products. Ice crystals grow rapidly at this temperature, causing protein denaturation and internal structure damage. As a representative of traditional freezing, it's used to contrast the advantages of liquid nitrogen freezing. -50°C is chosen to explore the nonlinear impact of temperature on ice crystal formation and verify the assumption that "lower temperatures provide better protection. -80°C tests the hypothesis that quickly passing through the maximum ice crystal formation zone (-1°C to -5°C) inhibits ice crystal growth, minimizing mechanical damage. International food freezing standards suggest that conventional freezing, as recommended by the International Institute of Refrigeration (IIR), should be at ≤-18 °C for aquatic products, close to -20 °C, but with larger ice crystal sizes (about 50-100 μm). For fast freezing technology, the U.S. Food and Drug Administration (FDA) requires individual quick freezing (IQF) at ≤-35 °C, controlling ice crystal size to 10-30 μm, still less effective than LNF.
Comments 5: Line 104-113: Foaming properties are described, but the homogenization speed/duration is omitted. Specify these to ensure consistency in foam formation.
Response 5: Thanks for your suggestion. We revised the details about homogenization process.
Lines 114-115: The MP solution was diluted to 2 mg/mL and homogenized (12000 rpm,1 min).
Comments 6: Line 137-145: SDS-PAGE methodology lacks gel concentration (% acrylamide) and marker details. Include these for clarity and reproducibility.
Response 6: Thanks for your comment. We have revised the details about SDS-PAGE methodology.
Lines 149-151: The resolving gel contains 12% acrylamide, and the stacking gel 5%. PageRuler Prestained Protein Ladder (Molecular weight range: 10-250 kDa) serves as the molecular weight marker.
Reference: [20] Laemmli U.K. Cleavage of structural proteins during the assembly of the head of bacteriophage T4. Nature 1970, 227, 680-685.
Comments 7: Line 211-226: Solubility results are discussed, but statistical significance markers (e.g., p-values) are missing in Figure 1A. Add these to support claims.
Response 7: In figure 1A, distinct letters in the same column signify a significant difference (p < 0.05). Lowercase letters and uppercase letters represented statistical significance of freezing methods and freeze-thaw cycles, respectively.
Comments 8: Line 256-264: Figure 1 captions should explicitly state the number of replicates (n) for each data point to assess reliability.
Response 8: We have stated the number of replicates in data analysis (section 2.8).
Line 217-218: All experiments were performed in triplicate, and data were reported as mean ± standard deviation (SD).
Comments 9: Line 340-355: Table 1 notes "Distinct letters" for significance but does not explain the statistical test used (e.g., Duncan’s, Tukey’s). Clarify in the caption.
Response 9: Thanks for your helpful comments, and we have revised the statement.
Lines 369-372: In the same column, different lowercase letters indicate significant differences among freezing methods, and different uppercase letters indicate significant differences among freeze-thaw cycles, as determined by Duncan's multiple range test (p < 0.05). Data are presented as mean ± standard deviation.
Comments 10: Line 363-377: Surface hydrophobicity results (Figure 6A) should correlate with fluorescence data (Figure 6B-D). Discuss this relationship explicitly.
Response 10: Thanks for your helpful suggestion. We added the discussion about their relationship.
Lines 419-421: The variation trend of fluorescence intensity is in line with that of surface hydrophobicity, indicating that more hydrophobic groups were exposed with the increase of FT cycles.
Lines 426-428: The results of surface hydrophobicity and fluorescence intensity indicated that LNF can delay the degree of denaturation of MP.
Comments 11: Line 404-413: The conclusion states LNF "significantly alleviated" MP changes but does not quantify "significant." Define effect sizes or percent improvements.
Response 11: In the conclusion, we comprehensively analyzed the protective effect of liquid nitrogen freezing on crayfish MP based on all the data. The quantified values were discussed in the preceding text using significance analysis.
Comments 12: Line 414-420: The funding statement mentions a grant but does not link it to specific experiments or outcomes. Clarify the grant’s role in the study.
Response 12: This research was funded by grants from “The 13th Five-Year Plan” National Key R&D Program “Key Food Safety Technology R&D” Key Special Project (Integration and Demonstration of Food Quality and Safety Assurance Technologies for the Whole - Industry Chain of Procambarus clarkii in the Middle and Lower Reaches of the Yangtze River, 2019YFC1606000). This study focuses on the quality changes of crayfish during transportation and storage, as well as the improvement methods.
Comments 13: Line 45-46: "temperature fluctuation would inevitably occur" → "temperature fluctuations inevitably occur" (simplify tense).
Response 13: Thanks for your valuable comments. We revised this sentence.
Line 50: Nevertheless, due to the time required to market these frozen crayfish, temperature fluctuations (-18 °C to -5 °C) inevitably occur during transportation and storage.
Comments 14: Line 104-105: "Foaming properties and foaming stability were analyzed following the method of Sharifian et al.[16] with modifications" → Specify modifications for clarity.
Response 14: Thanks for your helpful suggestion. We're sorry for the omission, and the missing word has been added to the text.
Lines 106-107: Foaming properties and foaming stability were analyzed following the method of Sharifian et al.[16] with slight modifications.
Comments 15: Line 152-154: The equation’s description is fragmented. Revise for fluency: "where A, A₀, A₁, and A₂ represent the absorbance values of..."
Response 15: Thanks for your helpful suggestion. We have improved this sentence.
Lines 162-163: Where A represents the absorbance values of the test group; A2, A1 and A0 refer to the absorbance values of the blank, standard and control, respectively;
Comments 16: Line 256-264: Figure 1 caption uses inconsistent formatting (e.g., "n FT"). Standardize to "nFT" or "n-FT."
Response 16: Thanks for your valuable comments. We reviewed the manuscript and unified the express.
Comment 17: Line 52: "cytoclasis" is uncommon. Replace with "cell rupture" or "lysis."
Response 17: Thanks for reviewer’s helpful suggestion. We have revised this word.
Comment 18: Line 363: "T-SH" is undefined. Spell out "total sulfhydryl groups" at first use.
Response 18: Thanks for your helpful suggestion. We have defined total sulfhydryl at the first use.
Line 377: As showed in Figure 5, the total sulfhydryl (T-SH) of MP was significantly decreased.
Comment 19: Avoid passive voice where active is clearer (e.g., "It was reported that..." → "Studies indicate...").
Response 19: Thanks for your valuable comments. We reviewed the manuscript and revised the necessary changes to the relevant sentences.
Comment 20: Define abbreviations (e.g., "MP") at first use in the abstract and main text.
Response 20: Thanks for your suggestion. We reviewed the manuscript and revised them.
Reviewer 3 Report
Comments and Suggestions for Authors
Very interesting work with relevant methods used to explore effect of liquid nitrogen feezing on crayfish quality. Few queries are listed below:
- Line 58-60, sentence incomplete.
- What is 30 in equation 2& 3. Give more details on method used for foaming such as volume, speed of homogenisation etc.
- Which kit was used for CA2+ ATPase activity
- Line 200- What is buffer B
- Are the units for y axis % in Fig 1 A. What about B? Pls mention in both.
- In Fig 2 why brackets have shading/texture?
- More discussion would improve quality of paper.
Comments on the Quality of English Language
Some rephrasing is required.
Author Response
Comments 1: Line 58-60, sentence incomplete.
Response 1: Thanks for your valuable suggestion. We checked the integrity of the sentences and made the necessary revisions.
Lines 64-68: This study delves into how LNF affects the functional properties, microstructure, protein integrity and conformation of frozen crayfish MP during FT cycles. By analyzing changes in molecular structure of MP, it reveals mechanism of LNF in protecting frozen crayfish quality. These findings offer a theoretical foundation for aquatic product cryopreservation.
Comments 2: What is 30 in equation 2&3. Give more details on method used for foaming such as volume, speed of homogenisation etc.
Response 2: Thanks for your valuable suggestion. The volume of myofibrillar protein diluent added in the experiment was 30 mL. The foaming steps have been modified by referring to relevant literature.
Line 115:The MP solution was diluted to 2 mg/mL and homogenized (12000 rpm,1 min).
Comments 3: Which kit was used for CA2+ ATPase activity
Response 3: As shown in line 160: The Ca2+-ATPase activity was measured using a microplate assay kit (ultramicro Ca2+-ATPase Kit).
Comments 4: Line 200- What is buffer B
Response 4: Thanks for your valuable suggestion. We have elaborated on and corrected the extraction solution.
Lines 210-213: Weigh 2.4228 g of Tris and 44.73 g of KCl, dissolve in 800 mL of distilled water with stirring, adjust the pH to 7.0 with hydrochloric acid, transfer to a 1000 mL volumetric flask, and dilute to the mark with distilled water to prepare the extraction solution.
Comments 5: Are the units for y axis % in Fig 1 A. What about B? Pls mention in both.
Response 5: Thanks for your suggestion. We have modified the solubility graphs. The turbidity of myofibrillar protein solution is measured in absorbance (Abs), which is usually expressed in absorbance units (AU, Absorbance Units) without specific units.
Comments 6: In Fig 2 why brackets have shading/texture?
Response 6: We are sorry for the mistake and we have revised it.
Comments 7: More discussion would improve quality of paper.
Response 7: Thanks for your helpful comment. We reviewed the manuscript and added necessary discussions.
Round 2
Reviewer 1 Report
Comments and Suggestions for Authors
Below are comments that were not addressed in the previous review.
The document presented in the references section does not include a web access link.
Develop and include a comprehensive experimental diagram of the study conducted and include it in the document.
Author Response
Comments: The document presented in the references section does not include a web access link. Develop and include a comprehensive experimental diagram of the study conducted and include it in the document.
Response 1: Thank you for your valuable comments. We added web access link with every reference. The comprehensive experimental diagram of the study also added.

Reviewer 2 Report
Comments and Suggestions for Authors
No more comments are required.
Author Response
Thank you